# Contested Welfare: Migrant Organizations in Search of Their Role in the German Welfare State

**Eva Günzel [1,*], Ariana Kellmer [2,*], Ute Klammer [2] and Thorsten Schlee [2]**

[1]   Urban and Regional Studies, Faculty of Social Science, Ruhr-University Bochum, 44801 Bochum, Germany
[2]   Institute for Work, Skills and Training, University of Duisburg-Essen, 47057 Duisburg, Germany
*   Correspondence: eva.guenzel@rub.de (E.G.); ariana.kellmer@uni-due.de (A.K.)

**Abstract:** This article examines the role of migrant organizations (MOs) in the welfare state and reflects on the transformation and negotiation processes in the organization of social protection in a society that is increasingly characterized by various forms of cross-border mobility. The article first describes various transformation trends in German social policy by highlighting the activation policy and marketization of social services. This transformation concerns not only the formal (material) forms of social protection and the relationship between migration and social policy, but also the organization of social protection within the German welfare state. By analysing qualitative interviews with representatives of migrant organizations and welfare associations, we then show which roles are ascribed to MOs by other welfare actors in the context of social protection and how the MOs position themselves. We argue that these role ascriptions are part of a negotiation process that goes along with the transformation of the German welfare state. MOs are increasingly addressed in the context of integration policy, while at the same time they are becoming more professional and are claiming a stronger role in formal security services. The discussion of the changing role of MOs in the future organization of the welfare state also sheds light on the question of the successful adaptation of social services to the needs of migrants in general.

**Keywords:** migrant organizations; social protection of migrants; transformed welfare state; pluralization of social services

## 1. Introduction

This special issue focuses on the changing role of migrant organizations as providers of social protection for migrants in Germany. The interest in this topic is not self-evident. The German discourse on migrant organizations has evolved and transformed in recent years. After the initial discussion of whether migrant organizations have disintegrative effects and promote the formation of segregated communities, the integration benefits have been emphasized more and more in the last two decades (for a summary overview, see Klie 2022). This discourse can also be situated in the context of the transformation of the welfare state. Our paper investigates the increasing relevance of MOs to migrants' social protection and examines under what sociostructural and political conditions this increasing relevance of MOs obtains. We argue that the shifting role of migrant organizations in welfare state arrangements provides us with insights into the transformations and negotiation processes in the organization of social protection in a society that is increasingly characterized by various forms of cross-border mobility. The article sheds light on the role of migrant organizations in (formal and informal) arrangements of social protection.

The first part of the article describes various transformation trends in German social policy. This transformation concerns not only the formal (material) forms of social protection and the relationship between migration and social policy, but also the organization of social protection within the German welfare state. The new role of migrant organizations in the architecture of social protection is shaped by the rationalities and practices of an activating welfare state. In parallel with this, integration and migration policies have

changed, as have migration-related discourses. Migrant organizations are now approached as relevant actors in the integration discourse, while other providers of social services are increasingly taking migration-specific needs into consideration in their own work as well.

The second part of the article shows how migrant organizations and welfare associations position themselves in the negotiation of social protection. Migrant organizations have changed considerably since the first organizations of this type were founded during the so-called guest worker migration in the 1960s. For a long time, they were perceived as purely cultural associations, but they have increasingly responded to migrants' needs for social protection. Today, migrant organizations provide a wide range of services, for example, in the areas of education, care, counselling and labour market access (SVR 2020). However, access to regular funding by the welfare state is still severely limited, partly because of the very close partnership between the state and the six large welfare associations in Germany,[1] which have been, and continue to be, entrusted with a large part of the state's protection tasks by social services. Welfare associations are significantly involved in negotiations about the organization of the German welfare state. We show how they try to legitimize their powerful positions and how they position themselves towards migrant organizations as new actors. We also contrast the roles assigned to migrant organizations by politics and administration, as well as by the welfare associations, with the self-positioning of migrant organizations.

The article concludes with a discussion of the pluralization of social services and the negotiations regarding the role of migrant organizations in this field. We show how the ascriptions and self-positionings allow for conclusions to be drawn about the organization of social services and social protection (for migrants) in the German welfare state.

## 2. Social Protection and Migration in the Transformed Welfare State

The transformations of the welfare state cannot be attributed to a one-dimensional or linear direction of development (Seeleib-Kaiser 2008, p. 210). Instead, different, and even contradictory, developments are to be expected in a field of constant political negotiation. Although the Basic Law (the constitution of Germany) establishes the welfare state as one of four structural principles of the state, the meaning and relevance of this remains politically contested. The normative foundations and social policy goals are left to political negotiation processes and rationalities. We assume that the addressing and the role, as well as the possibilities and problems, of migrant organizations can be explained in the context of the now dominant principles of the transformed welfare state. In the following, we will focus on three aspects of welfare state change: (1) substantive social legislation, (2) changing perspectives on the management of migration in the welfare state and (3) the organization of social services and benefits.

### 2.1. The Transformation of Social Protection: Formal Social Policy

Since at least the early 2000s, various development trends have been captured by the global analysis of "activation policy" (Klammer et al. 2019, p. 6). Activation policy is characterized by at least two rationalities. The first of these rationalities echoes the long-standing liberal economic criticism of a "bloated" welfare state. Here, the widespread narrative is that the welfare state prevents the free development of competitive and market potentials in economies that are now aligning themselves globally. The welfare state is believed to be an obstacle to economic and individual development and is even suspected of creating disadvantages in global market competition. Concerns about Germany as a major centre of commerce and industry based on national economic arguments are linked to new normative foundations of social policy. It became increasingly apparent that the welfare state of the "Bonn Republic" (i.e., West Germany before the reunification) was oriented towards a model of male wage labour and a reproduced gender-based division of labour and inequalities (Dingeldey 2011).

Welfare state activities tend to be understood in terms of investment rather than in redistributive terms. Through preventive as well as compensatory social, family and edu-

cational policies, citizens are to be empowered to participate actively in market activities. The new welfare state seeks to promote the employability of citizens. Social assistance is regarded as social investment that promotes re-entry into the labour market. The focus is on particularly marginalized groups that are potential labour-market targets of activation policy—initially the unemployed and recipients of basic social protection benefits (Heidenreich and Rice 2016, p. 8; Bonoli and Natali 2011, p. 8; Hemerijck 2013). In this sense, MOs, as part of civil society, are increasingly addressed within the framework of activation policy by state authorities and other welfare actors. In their daily work, MOs provide formal social services and informal social protection generated within the organization through social network ties (Bonfert et al. 2022b; Günzel et al. 2022).

Accordingly, the core of welfare state reforms in this era was related to creating and improving the employability of groups that had fewer employment opportunities. The reforms of the late 1990s and early 2000s included the restriction of labour regulations and the restructuring of social policy benefits. Other aims of the reforms were to ease the burdens on companies by limiting non-wage labour costs (social insurance contributions), a more market-based organization of social protection and the establishment of competitive structures at the various levels of public service provision (Bäcker et al. 2020, p. 39). For this workfare regime, social policy is geared "to the demands of labour market flexibility and employability and to the demands of structural or systemic competitiveness" (Jessop 1999, p. 23).

"Activation", as used in this context, is an ambiguous concept. It means enabling individuals to make use of previously unused potentials, but also includes an authoritarian call to gain employability and take up employment as soon as possible. Since around 1995, the transformation of the social security system has been accompanied by an increase in precarious and low-paid work.

### 2.2. The Transformed Welfare State and Migration

This new political rationality also affected the approaches to migration. Migration discourses are inscribed in national historical narratives and make up specific national paradigms of dealing with migration (Thränhardt and Bommes 2010). For a long time, the German migration discourse was characterized by the counterfactual claim that the Federal Republic of Germany was not a country of immigration. This paradigm changed, not incidentally, in parallel with the transformations of the welfare state described above, the aim of which has been to mobilize groups of people previously excluded from the labour market to find gainful employment. At the same time, a discussion of the "demographic time bomb" (Auth and Holland-Cunz 2007) began: the financial viability of the social protection systems seemed to be questionable in the long run given the age distribution of the population. The different forms of cross-border mobilities were rationalized under the still-national lenses of demographic change and a "shortage of skilled labour". In this new architecture of a national social state, the relation between migration and social policy changed fundamentally.

The transformations of the welfare state were reflected in the social protection laws, especially in the newly introduced Book II of the German Social Code (SGB II). They were also reflected in reformed labour law and in a number of changes to citizenship and immigration law that were implemented at the same time. The culmination of this was the establishment of the Independent Commission on Immigration (also known as the Süssmuth Commission, so named after the erstwhile Federal Minister of Youth, Family, Women and Health, Rita Süssmuth), which formulated the paradigms of immigration policy in the context of demographic concerns (UKZu 2001).

The following reforms highlight these developments:

- The revision of German nationality law in 2000;
- Various attempts to encourage skilled-labour immigration from third countries, which began with a green card scheme in 1998 and led to various legislative initiatives (Graf 2022);

- Changes in asylum law, the aim of which was to accelerate "integration into the labour market" and which also opened side doors for rejected asylum seekers who now have to prove their integration achievements (educational success or integration into the labour market; see Kolb 2021; SVR 2021).

Above all, the opening of Europe's internal borders has fundamentally changed the relationship between migration and social policy. The disparities in prosperity within the European Union ensured sustained immigration from the various new member states of the European Union, initially primarily from Poland, and after 2013 from Romania and Bulgaria (Graf 2022). Immigration of European citizens goes along with annually growing numbers of foreign employees in Germany. With the distinction between EU citizens and third-country nationals, the differentiation between nationals and foreigners is also taking on a new shape. As a result, persons from the Schengen Area can easily enter Germany, live and work there, but have only limited access to the social protection systems, depending on their individual and familiar situation (Bruzelius et al. 2017). As activation policy creates pressure to integrate people into labour markets, this pressure intensifies in combination with precarious legal status and limited access to basic income and other benefits and services. In various Western European countries, these gaps in social protection for European citizens are becoming increasingly obvious and confront politicians as social problems (Lafleur and Stanek 2017; Lafleur and Vintila 2020; Manolova 2017; Manolova 2019). Meanwhile, Germany is one the five countries with the highest number of refugees worldwide (UNHCR 2022).

With the continuing growth in the number of immigrants and of refugees in particular, including workers, their children and other relatives residing in Germany, the challenges and needs for social protection are changing. Different life-course regimes make it difficult to adapt to the systems of education and the organization of labour (Bommes 2011, p. 245). Specific needs, such as advice on the right of residence, on the transfer of qualifications into the German vocational training system, on experiences of racism and discrimination, on language-learning opportunities and on access to social benefits in view of the high level of legal complexity in the combination of residence and social law, lead to a continuing need for advice and support for migrant population groups who have to navigate the German welfare state.

Conditional and limited access to social services and benefits, combined with the structures of a flexible labour market that support precarious work, provide overlapping conditions of precariousness in terms of residence rights, employment, social protection and housing (Birke 2022). Against this background, migrant organizations are not only crucial actors in gaining access to welfare, they are also political actors that articulate migrants' interest in decent work conditions (Riedner 2018, p. 101).

### 2.3. Emergence of a Field of Integration Policy

It was not until 2005 that federal German policy responded to these newly emerging challenges with an emerging field of integration policy (Blätte 2015; Gestmann and Hilz 2017). This included, for example, counselling services for young people under the age of 25 ("youth migration services") and led to the introduction of nationwide mandatory language learning programmes, which were soon to become known as "integration courses". The federal government explicitly demands and promotes integration (§43 AufenthG), thereby using the semantics of activating social policies.

In Germany, it is mainly the federal states that promote comprehensive integration policies, depending on their political orientation and priorities. This is carried out in part through integration laws; the establishment of integration commissioners; and/or independent, though sometimes only temporarily implemented, integration programmes. In this field of integration policy, which has grown significantly in recent years and the aim of which is to address the specific social needs of migrants, the welfare associations have established themselves as the most important providers of social services for migrants (see discussion below). Likewise, migrant organizations are addressed by policy in this field

and increasingly appear as providers not only of migration-specific services but of other social services as well.

If we consider migrant organizations in the context of the transformation of the welfare state, we must also take into account the changes in the organization and management of social services.

### 2.4. The Organization of Social Services in the Transformed Welfare State

In addition to the formal legislation described above—which essentially concerns European Union law, federal German social law and the German Residence Act—the organization and processes of governing social policy have also shifted. In this section, we focus on this second level of governance and provision of social services because migrant organizations have increasingly found a new position within the structure of the welfare state in recent years (Pries 2022).

The organization of social services developed within nation state frameworks and path dependencies. The institutions of social welfare provision are set up mainly through national legislation and process control (Bode 2013, p. 28). Efforts to transnationalize formal social protection are in their infancy but are now gaining momentum, especially in European social law (Garben 2019). However, even in the few areas where European law intervenes in the national organization of social protection, implementation remains at the national level. This can be seen, for example, in the common European asylum policy, which is implemented very differently at each national level (Velluti 2016).

We therefore have to take into account the specific path dependency of organizing welfare in Germany. The German welfare state combines two different rationalities. During the uprooting and mobilizations of the early industrial age, social services were provided by local administrations (*Polizey*) and the churches, such as poor relief for mobilized people who were losing their economic subsistence. This local provision of welfare was superimposed by the social security system that was implemented by the nation state. It became obvious that the exclusion risks of an emerging industrial modernity could no longer be borne by local structures (Offe 1984). Both approaches characterize the dual-structure welfare landscape in Germany.

Welfare associations were founded to become part of civil society primarily in the context of local care for the poor. Over time, a corporatist model of governance evolved at the local level. The responsibility for social service provision is divided between local governments and non-profit organizations that are organized in the six large welfare associations (*Wohlfahrtsverbände*). They cooperate in the provision of public social services. The legislative bodies have a funding obligation for all the social services they wish to provide. In accordance with the principle of subsidiarity, the state does not provide these services itself if non-state actors (e.g., churches, welfare associations, private organizations) can provide them. This gives the welfare associations privileged access to the provision of local welfare. At the same time, they have their position in the local decision-making bodies that decide on the design and awarding of social services. This grants them both privileges and political influence. This system of social service provision has been a major characteristic of the dual-structure welfare state for decades (cf. Tennstedt 1992, p. 342; Grohs et al. 2017, p. 2577; Olk and Pabst 1996).

The transformation of the welfare state has also affected the organization of social services and thus of the welfare associations as the most important providers.

### 2.5. Quantitative Growth of Social Services

The transformation of the welfare state has not led to a dismantling of social services. On the contrary, the social sector is an economically significant growth market. In many German cities, welfare associations are among the largest employers. Although the figures should be used with caution, because the "third sector" has blurred edges and organizations, the growth figures of the welfare associations speak for themselves. While in 1990 they employed 750,000 people, by 2021 this number had increased to 2.1 million (BAGFW 2018;

Rauschenbach et al. 2021, p. 377). Still, Germany currently faces a shortage of skilled workers in the social sector.

In view of the ongoing austerity policies and the still widespread "withdrawal of the state" rhetoric, this development is, at first glance, surprising. After all, regardless of the social discourse and politically formulated austerity dictates, the number of employees in welfare associations has been growing continuously over the last 30 years. This indicates that activities of social reproduction, which used to be performed in family contexts, have become public affairs and thereby political affairs. This creates legal entitlements to social services and thus also new fields of work for welfare associations. In this context, services and institutions have become legalized, monetized, bureaucratized and professionalized (Rauschenbach et al. 2021). This, however, has made the organizations' relationship with their milieus—whether religious or, as in the case of the Arbeiterwohlfahrt (AWO), class-based—increasingly precarious. Now it is no longer the organization of the articulation of interests or the organization of a specific milieu, but the provision of social services that has become the central function of the welfare associations. As public contractors, they, too, create precarious employment conditions, and it has become difficult to distinguish these organizations from other competitors in the welfare markets (Möhring-Hesse 2018, p. 60; Klenk 2015, pp. 145–49).

The continued growth of the third sector has been accompanied by a transformation of the governance mechanisms of welfare provision, which is usually referred to as the marketization of social services. The public sector creates quasi-markets of social service provision. Following Ledoux et al. (2021, p. 9), we understand welfare markets as "politically shaped, regulated and state supported markets, which provide social goods and services through the competitive activities of non-state actors".

The marketization of social services has different dimensions and time layers. As early as the 1980s, instruments of New Public Management were increasingly introduced both in public administration and in the third sector (Nullmeier 2004, p. 496). These instruments led to product definitions for the services of public administration, different forms of incentive-creating and performance-related remuneration, and contract management within the administration and between the administration and the public. They have been shaping artificial "quasi-markets". The introduction of market mechanisms has seen different speeds and shapes in different sectors. Market-based steering mechanisms appear to be particularly advanced in the provision of labour-market-related social services and in the care sector. For example, most of the activation measures launched by local employment services are not provided by large welfare associations but by other private-sector or non-profit providers. This is a result of the contracting practices and the salary structures in this segment, which are not attractive to the welfare associations and their staff. Advanced marketization has led to wage and cost pressure, downward spirals for the employees of social service providers and, not least because of frequently changing sponsorship and inconsistent staff structures, to service quality risks (Knuth 2018; Greer et al. 2017).

Conversely, the field of social services has been broadened, and new providers and services have increased the choice of clients (Klenk and Nullmeier 2004, pp. 79–80). According to public choice theory, people should have the freedom and autonomy to choose the social services that suit them (Klammer et al. 2019, p. 6). This requires a pluralization of services: the marketization and pluralization of social services are the double-faced outcomes of welfare state reforms. Especially against the background of growing migration-related diversity in society, migrant organizations are now taking on new roles in various areas of these social services.

## 3. Welfare Associations and Migrant Organizations: Dynamics in the Transformed Welfare State

The realm of migrant organizations in Germany is diverse and dynamic. There are relatively well-established older organizations that were founded by "guest worker" migrants in the 1960s, and there has been a growing number of new organizations, especially

since the increase in refugee migration to Germany that began in 2015. Estimates of the number of MOs in Germany vary between 12,000 and 14,000, with some reaching as high as 17,000 (Priemer et al. 2017, p. 41; SVR 2020, p. 15). MOs are usually organized in registered associations (*eingetragene Vereine* or *e. V.* for short) but may also have a different legal status, for example, that of a non-profit enterprise. In addition, there are many informal initiatives or associations whose number exceeds these estimates, which only include registered associations. Against the background of the transformation of the welfare state, a new role for MOs in the provision of social protection is required both politically and by migrant organizations (SVR 2020; Halm et al. 2020). Migrant organizations are increasingly articulating their demands for participation and their desire for appropriate funding of their services, especially in the area of migration-related social work as it has developed within the field of integration policies. As new actors in the provision of social services, they operate in the same field as the "old" welfare associations, which are now struggling with the challenges posed by the changes in the welfare state. In the following, we will present empirical results that reflect the negotiation processes between welfare associations and migrant organizations against the background of the pluralizing welfare landscape. First, we will show the effects of transformation on the self-positioning of welfare associations and how this influences their positioning towards MOs. We will then contrast the welfare associations' role attribution towards MOs with the self-positioning of migrant organizations.

For that we use interview data and document analysis that were conducted within the framework of the research project "Migrant Organizations and the Co-Production of Social Protection" (MIKOSS) (2020–2023), a cooperative project of the Universities of Duisburg-Essen, Bochum and Dortmund. Our analysis is based on expert interviews with representatives of migrant organizations (n = 15) from three large cities in the Ruhr region, as well as expert interviews with employees of various welfare associations (n = 9) at the municipal, state and national levels. We applied coding and content analysis to the interview data. This was supplemented by interviews with employees of the municipal and state administration (n = 9) and by document analysis—in particular, documents on the state funding structures for MOs. In the following section, we will present selected empirical results, with a special focus on the effects of welfare state change and the resulting tensions in migration-related social work, in the context of which both welfare associations and migrant organizations (re-)locate themselves.

*3.1. Effects of Welfare State Change on Welfare Associations and Their Positioning towards MOs*

In the expert interviews, the representatives of the welfare associations were asked about the role they ascribe to migrant organizations in the field of social services and how they themselves (co-)organize social protection for migrants. It became clear that the argumentation and self-positioning of the welfare associations are strongly influenced by the transformation of the welfare state, about which they are mostly sceptical. They are struggling with the increased influence of the state on the shaping of services and the weakening of the principle of subsidiarity, and they want to distinguish themselves from new actors. The roles they ascribe to migrant organizations must be understood against the background that the traditional welfare associations must defend their role in the welfare state. In the course of the transformation of the welfare state, the state has come to exert more control. Welfare associations are now less free to act and are more accountable for the state funds they use as providers of social services. In the course of marketization, they are increasingly under pressure to work efficiently, which has led to a deterioration in working conditions in some areas. As a result, welfare associations are struggling for legitimacy, for example, when they advocate good working conditions for the disadvantaged, and at the same time, they must respond to the increased cost pressures in their role as employers.

The welfare associations react to these changes in different ways. In some areas, such as care, they have more or less bowed to the pressure to marketize. In others, they have withdrawn and decided to no longer become involved with certain conditions or

respond to certain tenders (e.g., EU projects for which administration costs would be too high). In the field of migration-related social work, which had not been particularly marketized in the past, welfare associations fight for the preservation of the principle of subsidiarity, their autonomy in the provision of social services and their influential position in the welfare landscape. To this end, they make use of their still strong connections to politics and administration and form *Ligen* ("leagues", i.e., working groups of the umbrella organizations of *Freie Wohlfahrtspflege*) in order to appear "together, with one voice" (Welfare Association X).

In the interviews, the close relationship between the welfare associations and the closed corporatist negotiations with politicians and administration were emphasized in a very similar way at all levels of the various associations (municipal, supraregional, state and federal levels). Welfare associations legitimize their powerful positions, which are closely tied to the state, and thus fight against the increasing influence of other actors and the pluralization of the landscape of actors. The representatives we interviewed do this by referring to a "win–win relationship", according to which the state benefits from (1) their expertise in the field, (2) their proximity to the clients, (3) the relationship of trust, (4) financial in-house contributions that they have to make in taking over the sponsorship and by which they also assume financial responsibility and (5) the quality of the services, which, in their opinion, distinguishes their own performance from that of newer and purely economically oriented actors.

The welfare associations emphasize, in particular, the trusting relationship and their own reliability, and they distance themselves from new economically oriented actors, whom they regard as unreliable in the long term, both in terms of their organizational structures and the services they offer. They also distance themselves from some NGOs and purely interest groups, which could be more radical in their demands and do not take state interests and resources into account.

Representatives of welfare associations criticize not only the marketization of social services, in which they observe both a loss of service quality and deteriorating working conditions for employees. They are also sceptical about the growing influence of local, state and federal governments on the design of social services, especially in cases where local governments are increasingly assuming responsibilities themselves. However, they rarely put forward economic arguments against an expansion of state control. Rather, they struggle for social influence and (political) co-determination in the shaping of the welfare state.

The above-mentioned challenges that welfare associations face in the course of welfare state change also influence their positioning towards migrant organizations. The interviews we conducted with representatives of welfare associations revealed a complicated relationship between welfare associations and migrant organizations. In their role as sociopolitical actors, but also in their self-image as humanitarian and charitable organizations, welfare associations see themselves as advocates for migrants, and they welcome and support migrant self-organizations. However, they also compete with migrant organizations as social service providers for resources and clients.

Due to the changes in the welfare state, the welfare associations see the "principle of subsidiarity" as being at risk. In their struggle to preserve the principle of subsidiarity and defend it against further marketization, the welfare associations concede that the MOs are involved in shaping society and assuming social tasks. On the one hand, they argue for the participation of MOs in the creation of social protection. On the other hand, they are sceptical about migrant organizations as new actors in the provision of formal social services and thus as part of pluralization and marketization, although they do not always differentiate between social entrepreneurs, NGOs and MOs. They compete not only for contracts and funding but also in their claim to legitimacy in advocacy for migrants and for professional competencies in the care and counselling of migrants. German welfare associations typically offer their social services in the context of a broader claim to the shaping of society and social conditions (Möhring-Hesse 2018, p. 60).

The positioning of welfare associations towards MOs reflects the already mentioned pressure to legitimize themselves towards the state and the milieus they represent, as well as towards their own professional field (cf. Möhring-Hesse 2018, pp. 60–61). Thus, they legitimize their more powerful positions with their more pronounced professionalization. Although MOs increasingly offer professional social services, the employees of welfare associations regard MOs as "voluntary providers". From their point of view, MOs establish access to clients, acquire participants and are a "place" where clients can be found. They do important work when it comes to integration and orientation for people within the social protection system, but they do not (yet) see migrant organizations in the role of social service providers or even welfare providers. Firstly, they consider the range of services offered by MOs as (state-funded) social service providers to be small. They regard MOs primarily as providers of integration-related services and partly reproduce the integration-political addressing of MOs. Secondly, they have a critical view of the future role of MOs as welfare providers. In addition to the low degree of professionalization, the welfare associations justify their critical stance with reference to the post-migrant society and to the concern that MOs might predominantly represent particular or special interests and contribute to the formation of an alternative system of social protection that leads to segregation. At the same time, they emphasize their own claim to universal services—that is, their openness to every client group.

> "But if the local services are already so multilingual and achieve good results based on extensive experience, I don't see any need for further specialization in one migrant group. So, I think that's the special challenge for migrant organizations, as I said. What do they want? Do they want advocacy for a specific clientele? That would be as if we were there exclusively for Christians. And I believe that we have long since passed that stage, so to speak". (Welfare Association, federal level)

The religious organizations continue to be oriented towards the churches and Christian values. However, in terms of their clients, they see themselves as being open to all people and as advocates for minorities and people whose interests they believe to be under-represented in society. Other welfare associations emphasize their historically grown responsibility for certain groups of migrants and the comparatively high proportions of employees with a migrant background within their workforces.

The welfare associations not only seek to distinguish themselves from migrant organizations; they also struggle with the legitimate role of migrant organizations. As actors in "integration", MOs are very welcome, and they are also welcome when it comes to active participation in "city society" and jointly resolving social problems. However, welfare associations are critical of MOs as professional providers of social services, especially when the MOs are perceived as "social entrepreneurs" or economically oriented actors.

In addition, welfare associations find it difficult to integrate migrant organizations into the German welfare system. MOs should integrate into the welfare structures in a way that does not jeopardize the existing corporatism and the principle of subsidiarity, for example, by joining a welfare association (e.g., Der Paritätische) or by founding their own umbrella association and thus being available as contact persons.

> "I find it unquestionably right and legitimate that migrant organizations move and participate in the social market and try to make offers—no question at all. The question is how the competition, the togetherness, is then organized, and according to what conditions". (Welfare Association, federal level)

The main problem welfare associations see in this context is not the legitimacy of migrant organizations as such but the organization of the welfare state. Some projects initiated or supported by welfare associations aim at the professionalization of migrant organizations, and the establishment of a Muslim welfare association has been discussed as well. This is another expression of the struggle for the inclusion of the MOs and an attempt not to put the existing order at risk.

*3.2. The Self-Positioning of Migrant Organizations and Their Relationship with Welfare Associations*

As discussed in the previous section, the pluralization of the landscape of actors as part of the transformation of the welfare state has led to tensions, with welfare associations trying to keep their historically evolved position and migrant organizations repositioning themselves as relatively new actors. Locating migrant organizations within the welfare landscape is a complex task, primarily because they are difficult to grasp as organizations. Migrant organizations encompass a wide range of organizations with (1) different identity-generating contexts; these may be national or regional, religious, ideational or issue-/task-based references. They also exhibit (2) different degrees of professionalization, and (3) the migration reference is present in the organizations in different forms. What constitutes a "migrant organization" always depends on the definition of "migrants". Accordingly, the term "migrant organization" refers to a variety of heterogeneous organizations with different claims to be located in the welfare landscape.

Regarding the role of migrant organizations with respect to migrants and their members' social protection from different perspectives, we can contrast the approach to the role of welfare associations described above with those of representatives of migrant organizations that we interviewed about their social-protection-related activities during our research.

Our field research took place between January and November 2021. We conducted 15 semi-structured expert interviews (cf. Helfferich 2019) with MO representatives. Due to difficulties in field access because of COVID-19-related restrictions, we contacted the MOs through gatekeepers, including employees of municipal integration centres who stay in regular contact with many migrant organizations or at least know about the field of MOs in their city. While structuring our sample, we used the theoretical sampling technique of choosing a range of organizations that would be as diverse as possible. We interviewed organizations with various target groups, among which are a group of migrants from a specific region, religious communities, multicultural organizations and associations only for women. Besides their diverse target groups, the MOs range from entirely voluntarily organized and rather small associations to highly professionalized organizations that employ staff and have offices in various locations. The funding sources range from only membership fees and donations to regular state funding.[2]

In the interviews, we asked the representatives of the MOs about the goals, organizational structure, funding sources and working methods of their organization; their activities in relation to social protection; and the challenges they see for migrants in this context but also in their own work. In addition, we surveyed their opinions on the coordination of social protection in relation to other players in the German welfare landscape and about their specific contacts and relationships with other actors, such as state institutions, welfare associations, and other civic or private organizations.

Generally speaking, migrant organizations are aware that their work is indispensable for the social protection of migrants. Within the MOs' structures, and encouraged by the organizations, informal social protection is achieved through personal connections and the exchange of private concerns. This is a phenomenon confirmed by many interviews in which the MO representatives highlighted the familial atmospheres in their organizations, which encourage their members to speak openly about their concerns and wishes. In this sense, through this form of exchange and collective problem solving, MOs create social protection incidentally. We use the term "*homemaking*" to refer to this rather unique function that MOs fulfil in the context of social protection and which occurs less typically in professional welfare associations (see Bonfert et al. 2022a in this Special Issue). At the same time, MOs offer many social services similar to those provided by other welfare actors. These include formal offers, such as German language courses, consultation services and workshops on issues related to labour, health, education and other areas.

Thanks to their proximity to their target groups, they are often the first point of contact for migrants with concerns of any kind. As associations, they grow organically with the

demands placed on them by their members and usually without clearly regulated organizational structures and departments, although migrant organizations vary considerably in their degrees of professionalization, and there are organizations that work in ways quite similar to professional welfare associations. In response to weak financial capabilities and a lack of resources, MOs have built a broad network of contacts over the years without whom they would struggle to manage their daily—mostly voluntary—work.

Regarding their relationship with welfare associations, the analysis of the egocentric organizational networks[3] of MOs that we surveyed as part of our research revealed that, in their daily work, migrant organizations are embedded in a dense network of mostly local contacts with various civil-society and government actors and organizations. The networks were analyzed according to type of actor, their relevance to the MOs and type of relation (neutral, positive or negative; see Günzel et al. (2022) in this Special Issue). These ties include forms of cooperation, (financial) dependencies or even conflictual relations, as well as loose contacts that the MOs use to exchange information and other forms of mutual support. In these networks, welfare associations play a central role for most of the interviewed MOs. They are either members of a welfare association themselves as an association, share staff and spatial resources, or set up joint projects. Of the 15 MOs surveyed, 14 have regular contacts with one or more welfare associations, these being the third most frequent network contacts of the MOs after other migrant organizations and state actors and institutions. The MOs also rate welfare associations as quite important partners, especially when it comes to access to funding programmes and professional advice for migrant organizations.

Nevertheless, MOs also perceive tensions between themselves and welfare associations. One reason for these tensions is that competitive situations can arise. A representative of an MO describes the position of his association as a "pass-through", in the sense that the MO trains personnel, for example, in intercultural education expertise, and the employees then move on to large welfare associations that offer better salaries or the prospect of long-term employment. As less well-established players with poorer financial resources than the large welfare associations, the drain of well-trained staff results in a loss of both human resources and expertise, which places an additional burden on MOs:

> "We have a lot of people who stay for one or two projects and then say, Hey, it's been nice with you, but now I need a different challenge or more protection. And then, yes, they leave (laughs softly), which is a shame for us, of course, because that's always a loss in terms of knowledge. That's for sure. We also train our employees. They gain a lot of knowledge from us. And that is then lost for us. We lose that experience". (Dorado International e. V.)

Two MOs from our sample that have increasingly professionalized their services in recent years reported that they also compete with welfare associations for funding and that they often lose bids for funding because they lack the expertise, time and staff required to deal with the sometimes-complicated application procedures. In addition to this competitive situation, there are dependencies between welfare associations and migrant organizations:

> "As I said, the municipalities, but also the federal government, they mainly have the welfare associations as contact persons, a result of which is that the migrant organizations tend to join these welfare associations or have to join the welfare associations, whether they want to or not. If they want to have opportunities, financial resources, they have no other options than to say, Hey, I'm going to AWO, I'm going to the Paritätische, yes. But I have also experienced a lot of the opposite. So, I experience a lot that welfare associations suck their clientele out of the migrant organizations, but don't let the migrant organizations participate". (Lomingo e. V.)

Another reason for the tensions between MOs and traditional welfare associations is the multidimensional power imbalance that becomes visible in this context. On the one hand, MOs perceive membership in and cooperation with welfare associations as an

opportunity to access funding and participate at the municipal level, and they are thus forced to maintain (good) relations with the welfare associations and eventually become members. On the other hand, MOs feel exploited as "suppliers" of clients for the services of the welfare associations. MOs often report that large welfare associations fill their lists of beneficiaries of their welfare programmes with the help of MOs and their contacts, who in return are not adequately appreciated:

> *"That means the migrant organizations always have to fight for survival because they don't have the financial resources, and at the same time, they still serve the wishes of the other welfare associations that get all the money"*. (Lomingo e. V.)

At this point, instead of their role as suppliers, many MOs hope for better opportunities to shape the municipal structures of social protection for migrants, as well as for better funding conditions for their own programmes. Their expertise in migration-related issues, their intercultural knowledge and their proximity to the target group could enable MOs to meaningfully contribute to the development of support structures for migrants if they were offered better opportunities for participation.

As discussed in the previous section, welfare associations sometimes accuse migrant organizations of not working professionally enough and of acting in an exclusionary manner towards other population groups by providing services for only one particular target group. However, in a survey conducted by The Expert Council on Integration and Migration (SVR), 76% of the MOs stated that their services are used by people with a migration background—regardless of their region of origin—with 43% stating that people with no migration background also use their services. These figures are in line with our own research: a majority, especially those that have become highly professionalized and have grown over the past few years, offer their services to an intercultural audience that includes people with no migration background. The main exceptions here are religious communities, although it was also emphasized that, in general, everyone is welcome. The accusation of offering services to only one specific group of people also appears to be answered by the MOs studied, since many of them have strongly emphasized their openness to any group or person, both in our interviews and in their public communications and on their webpages.

Regarding professionalization, our results show a heterogeneous picture: our sample includes associations that are managed entirely through volunteer work, as well as larger migrant organizations whose staffs consist entirely or primarily of permanent employees and honorary staff. Most MOs also offer formal services (e.g., German language or other professional training courses and fixed advisory structures) and informal types of social protection (e.g., empowerment and advocacy, crisis aid, or informal get-togethers where the members can discuss their concerns). In particular, the larger associations are divided into work units, each with their respective independent organs. From these structural features—the relationship between full-time and voluntary work, the formality of the services offered and the organizational structure—we can conclude that many MOs do indeed act as professional service providers (see Bonfert et al. 2022a and Günzel et al. 2022 in this Special Issue). One difference from welfare associations, however, is that some MOs do not want to act as official professional welfare providers, although some already are and their services are equally important as those of established welfare associations. MOs tailor the services they offer to the needs of their target groups, thereby continuing to grow over the years as a result of the high demand for their work. What they instead wish is to be adequately valued for the work they already do. Rather than being competitors of the major welfare associations, MOs want to be accepted as partners of welfare associations who not only provide their expertise in integration policy issues on demand but can also actively help to shape them.

## 4. Contested Welfare: Migrant Organizations in the Transformed Welfare State

The positioning of migrant organizations and welfare associations against the background of a transformed welfare state opens up a dynamic field of negotiations about the

organization and provision of welfare. Based on our empirical research, we can identify different lines of conflict around the continuous shaping of the welfare state.

### 4.1. Questions of Identity

There is no general answer to the question of what a migrant organization is. As Fauser (2010, p. 285) notes, this is due to the vague meaning of the two components of the term "migrant organization". It is not only the growing doubt about the term "migration" or the question of "how long a person will be a migrant". Some scholars (e.g., Faist 2013; Bruzelius and Shutes 2022) tend to observe mobilities instead, because observing migration is always nation-state-centred. In addition, some organizations do not want to be addressed as migrant organizations. Instead, some of them refer to themselves as "*Neue deutsche Organisationen*" (New German Organizations) because they do not want to be labelled as migrant organizations. At the same time, we observe various organizational forms that deal with the different social questions arising from mobilities.

Another identity-related issue is the tension between professionalization and association activity. An organization supported by professionals develops its own dynamics. The professionals must manage new project funding in what are often highly competitive environments. The organization has a responsibility to keep its staff employed and also has an interest in growth and building further resources. This creates tensions between the operation of the organization and the day-to-day lives of voluntary members of the association, as well as with the political function of representing interests, as we can also see from the large welfare associations losing their identity and their ground. Is there any relation remaining between the *Arbeiterwohlfahrt* and the wage-dependent employees? In dealing with the structures of organizing welfare in the transformed welfare state, the professionalized migrant organizations face the same structural problems as the established welfare associations. Still, migrant organizations, both the more professionalized and the less professionalized ones, emphasize their bottom-up approach (cf. Halm et al. 2020), as they claim to always adjust their offers to the needs of their target groups.

### 4.2. Migrant Organizations between Social Services and Entrepreneurship

Regarding financial resources and structural positioning, we find that larger and more professionalized migrant organizations would like to receive regular state funding and have access to social service providers, whereas smaller and volunteer-based associations are less likely to perceive themselves as welfare providers. However, they would still like their work to be valued and funded.

Since MOs vary greatly in their size, range of tasks and target groups, the roles that state actors and welfare associations ascribe to MOs vary as well, and it is not always clear what type of organization is being referred to. As a result, MOs can be understood as NGOs, social entrepreneurs, cultural associations or professional service providers. This heterogeneity in the ways they are perceived is related to the pluralized landscape of actors and to the nesting of tasks, functions and memberships of migrant organizations.

> "*A current trend can be observed in recent years among migrant organizations: There are some that work professionally, also registered associations because they have discovered this area for themselves and know that they can earn money through corresponding funding. There are some who make good money and have discovered this. Well, you have to differentiate: These are organizations that have come into being because they say, 'We want to make offers and earn money.' Then there are others who say, 'We are idealists.' There are pigeon fanciers or whatever, 'We just want to do that,' or there are local associations, there are especially many Turkish ones. As I said, there are very different nations, cultural backgrounds*". (Local Integration Administration)

Above all, the increased competition from a growing number of "social entrepreneurs" is viewed critically. Here, too, however, it is difficult to draw a line. "Social" entrepreneurship can be defined as entrepreneurship that has social relevance both in terms of its range of tasks and its impact on society. In this respect, the scope of activities should include

services aimed at people in special problem situations or with special assistance needs and should serve to reduce these problem situations and satisfy assistance needs (Heinze et al. 2011, p. 91). These organization can be small non-profit associations, but also large foundations or individual companies. The term "social entrepreneurs" may refer to for-profit or non-profit organizations, with "non-profit" not necessarily being synonymous with "public-welfare-oriented" (Heinze et al. 2011, p. 92).

### 4.3. Hybrid Organizations

In addition to the unclear attributions of organizational form, the different tasks, roles and memberships blur the lines between organizations. There are migrant organizations that are members of (municipal) umbrella organizations. These umbrella organizations develop professional structures to support their members, e.g. in articulating themselves at a political scale. However, in some cases, a member organization and its umbrella organization compete for the same funding and projects. At the same time, both smaller migrant organizations and their umbrella organization are members of a welfare association, the Deutscher Paritätischer Wohlfahrtsverband (Der Paritätische), which consists only of member organizations. Sometimes, it remains unclear whether an umbrella organization represents the interests of its member organizations or whether it competes with these organizations for funding. Both the Paritätische as a welfare association and the umbrella organization of migrant organizations, as well as the MOs, as members of the Paritätische, can act as providers of social services.

Migrant organizations usually perceive themselves as being autonomous in their role as social service providers. In contrast, representatives of welfare associations or politicians may address them as members of the Paritätische. Thus, if migrant organizations are not regarded as relevant providers of social services, this may be because they are assigned to the Paritätische and are thus regarded as a member of a welfare association. In some cases, they are not defined as migrant organizations because they are regarded as more financially oriented providers.

### 4.4. Target Group Orientation versus Universalistic Social Policy

The comparison of migrant organizations and welfare associations as social policy actors revealed tensions between target-group orientation and the opening of services "to all". The welfare associations, in particular, argue that their services are directed at every social target group. In contrast, in our research, the welfare associations were often perceived more as official state actors than as self-organizations, due to their proximity to the public sector and their high degree of professionalization. Behind the difference between target group orientation and universalistic social services are conflicting models of society and thus also conflicting policy models, a finding to which Scholten (2020) has recently drawn attention.

In their self-descriptions and in the descriptions of others, MOs open up access to communities and, through their close contact, promise a better fit of social services to the highly differentiated needs of super-diverse societies. This orientation towards very small target groups is in conflict with sociopolitical approaches that identify and address needs in a standardized and large-scale manner. Activation policy measures in the labour market as well as integration courses (and many other courses and programmes, perhaps even the education system in its entirety) usually follow standardized procedures and central plans. These measures "integrate" into a society that is clearly defined and that knows what it is and what everybody should be. They rely on standardized procedures and seek to ensure a uniform level of quality and teach comparable skills. These industrialized, classical, modern-age approaches no longer seem to be suitable for the conditions in liquid, super-diverse, complex societies (Bauman 2000; Vertovec 2007; Scholten 2020).

Cooperation with migrant organizations and other small-scale member associations in the provision of social services can point the way to new policy models that no longer start from state-led integration processes but describe their problems themselves in local actor

networks and try to find suitable solutions. This is no small task, as it requires thinking about standardized life courses, educational qualifications, and valuing different types of resources and attempts that rely on individual resources and potentials.

### 4.5. Pluralization and Marketization

Marketization and pressure to perform in the provision of social services push the large welfare associations into a position where they must defend their privileged position in the corporatist mode of organizing social services. The resulting tensions are currently moderated by differences in access to social services. Migrant organizations are mostly visible in the field of integration policy. However, they also complain about barriers to access in this field: their structures make it difficult for them to raise their own funds and succeed in competitive award procedures. In most cases, they are not involved in the negotiation processes for the design and allocation of social services. At the same time, there is the negative scenario of uninhibited competition among a large number of providers of social services, the result of which is a precarious labour market for social or care workers in which service providers must constantly seek to undercut one another and the quality of the social services is often subordinated to competition among organizations.

## 5. Conclusions

In this article, we have examined the role of migrant organizations in the social protection of migrants within the new architecture of the transformed welfare state in Germany. We have explored the sociostructural and political conditions in which migrant organizations have become increasingly important for the social protection of migrants. We have shown that the changing role of migrant organizations allows us to look at the changes and negotiation processes in the organization of social security in a society that is increasingly characterized by various forms of cross-border mobility.

In the context of a transformed welfare state, the relationship between migrant organizations and welfare associations points to central sociopolitical areas of tension and lines of conflict. The transformation of the welfare state has opened up the field, in which MOs can operate both formally and informally. One of its dubious effects is that it now reassesses migration in the welfare state and seeks to provide opportunities for migrants, while at the same time creating precarious residence rights and working conditions. It opens up the field of social services and promotes civic engagement, while also creating structures for ruinous competition among organizations with which both the new and the established social service providers are struggling. This development has transformed the relationship between the established welfare providers and state actors and has made welfare providers increasingly dependent on public policies.

In this context, we observed a broad field of migrant organizations (see also Günzel et al. (2022) in this Special Issue), such that it is difficult to grasp and describe these organizations as a collective group. This concerns the reference to "migration" as the doubtful unifying feature of these associations, but it also concerns the numerous types of organizations and areas of activity, which range from essentially voluntary activities to mixed forms to profit-oriented migrant entrepreneurship. The fact that voluntary organizations are also of central importance for informal social protection practices is presented in detail in this Special Issue (see Barglowski and Bonfert 2022). Often, these MOs do not wish to get involved in additional areas of activity but only to obtain better access to funding opportunities for their association activities without having to build new professional structures for this particular purpose.

If MOs also become professionally active as social service providers, they potentially compete not only with the established welfare associations but with other migrant organizations. This playing field is characterized either by corporatist negotiation systems or by competition. In the first case, the welfare associations have privileged access. They defend it with reference to harmful competitive effects and their function of representing interests. In sectors such as activating labour market services or in the health sector, on the other

hand, the public sector has set up precisely these competitive structures. This opens up access and a plurality of social services, but also leads to a subordination of social services to the requirement of having to survive in competition.

These tensions are not easy to resolve. Analytically, they are the dynamic factors in the reproduction of welfare. They take shape in everyday interactions and situations—in local and field-specific contracting practices as well as in the contact between organizations and members or clients. It is here that we see the potential of the increasing inclusion of migrant organizations in the provision of welfare. Due to their heterogeneity, they can enrich and thus improve the local design and provision of social services with their specific knowledge and experience, even beyond the integration expectations of the public sector.

In some respects, the German welfare state is different from other (European) welfare states, primarily because of its close partnership with the large welfare associations, which have been assigned a large number of state tasks. We have shown that this particular system of organizing welfare influences the role of migrant organizations and their opportunities to establish themselves as professional service providers. However, in their daily work, MOs address the needs of migrants straightforwardly and support people in establishing reliable access to social protection. Therefore, MOs should continuously be explored as an important component of societies that are constantly changing as a result of migration and mobility. An international comparison would be useful to understand how different welfare state arrangements influence the role of MOs in various countries.

Another aspect that requires further research is the different ways in which social services are allocated at the municipal level of an increasingly marketized welfare state. Our empirical material suggests that the role of migrant organizations also depends on differences in the allocation of social services at this level. Obviously, despite legal changes, municipalities have different degrees of latitude in organizing social protection services.

**Author Contributions:** Conceptualization, E.G., A.K., U.K. and T.S.; Data curation, E.G., A.K., U.K. and T.S.; Formal analysis, E.G., A.K., U.K. and T.S.; Funding acquisition, U.K. and T.S.; Investigation, E.G., A.K., U.K. and T.S.; Methodology, E.G., A.K., U.K. and T.S.; Project administration, U.K. and T.S.; Software, E.G., A.K., U.K. and T.S.; Supervision, E.G., A.K., U.K. and T.S.; Validation, E.G., A.K., U.K. and T.S.; Visualization E.G., A.K., U.K. and T.S.; Writing—original draft, E.G., A.K., U.K. and T.S.; Writing—review & editing, E.G., A.K., U.K. and T.S. All authors have read and agreed to the published version of the manuscript.

**Funding:** This research was funded by the Mercator Research Center Ruhr (MERCUR) (Grant number Pr-2019-0049). The APC was funded by the Ruhr-University Bochum.

**Institutional Review Board Statement:** The research study entitled "Migrant organizations and the co-production of social protection" underlying this article includes human research participants. The study was prospectively approved by the legal offices of the Ruhr-University Bochum and the University of Duisburg-Essen. Ethical approval was not mandatory for this study.

**Informed Consent Statement:** Informed consent was obtained from all subjects involved in the study.

**Data Availability Statement:** To protect the privacy of our research participants, research data are not shared.

**Acknowledgments:** We want to thank Karolina Barglowski, Lisa Bonfert, Sören Petermann and Ludger Pries for their help as well as all the individuals who supported us in our research and who shared their perspectives and personal stories with us. In addition, we are very grateful for all the feedback provided by our research team colleagues.

**Conflicts of Interest:** The authors declare no conflict of interest.

## Notes

[1] These six welfare associations are the Arbeiterwohlfahrt (AWO), the Deutscher Caritasverband (Caritas), the German Red Cross (DRK), the Diakonisches Werk der Evangelischen Kirche in Deutschland (Diakonie), the Deutscher Paritätischer Wohlfahrtsverband (Der Paritätische) and the Zentralwohlfahrtsstelle der Juden in Deutschland (ZWST).

[2] See Günzel et al. (2022) for a detailed overview of the MOs' key characteristics.

[3] During the interview the organizations were asked to mark all their contacts related to social protection on a network map consisting of concentric circles with the MO in the centre. The closer the actor was placed to the MO, the more important that actor was to their work. The connections could then also be labelled as neutral, positive or negative.

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
