# Peer review of "Contested Welfare: Migrant Organizations in Search of Their Role in the German Welfare State"

_socsci, doi:10.3390/socsci12020090_

Round 1
Reviewer 1 Report
Dear authors,
I read the article titled "Contested Welfare: Migrant Organizations in search of their role in the German Welfare State", which examines the role of migrant organizations for the social protection of migrants against the background of a transformed welfare state in Germany, and in relation with the central role of the larger welfare organizations.
I found the article interesting, as it sheds light on a specific country case study (Germany), while addressing issues and challenges of welfare systems that can be seen also in other countries across Europe. The context description is accurate, and enables a reader who is not familiar with the organization of the German welfare system to learn about its fundamental structure and understand the dynamic processes of its evolution described in the paper.
In the first part, the article analyzes the main transformations of the German welfare system, focusing on the role and positioning of welfare organizations. It then turns to the emerging role of migrant organizations, before presenting the empirical results of the interviews that the authors have conducted with representatives of both welfare and migrant organizations.
When describing the results, more details on the nature of the organizations reached through the interviews would be useful. In particular when dealing with migrant organizations, given the described heterogeneity of their nature and forms, the reader should be informed on which organizations have been interviewed (key characteristics such as size, scale of operations, services offered, identity/community, etc.). It would also be important to know which criteria were adopted in making the choice of which organizations to interview.
Specific comments:
- line 152 "Above all, the opening of Europe's internal borders, the relationship between migration and social policy have been changing fundamentally." This sentence is incomplete/unclear.
- line 260 "that indicate that activities of social reproduction, which were previously performed in family contexts, have become public and thereby political affairs." It is unclear how this sentence logically follows from the one before.
- line 308 "new organizations are emerge steadily" This sentence seems incorrect and therefore unclear.
Throughout the paper, there are some repetitions that could be addressed to improve readability. As a final remark, I recommend to carefully check punctuation, brackets, capital letters, numbering of paragraphs, etc. Some editing for English language is also recommended.
Author Response
We would like to express our gratitude for the thoughtful and appreciative feedback. In the following, we explicate the amendments we have made with reference to the reviews. Overall, our article has gone through an English proofreading process. In addition, we address all reviewer comments and revision notes in the table below.
REVIEW 1 |
|
Revision recommendation |
Revision |
Info on migrant organizations is missing: what are the characteristics of the organizations and how were they selected?
When describing the results, more details on the nature of the organizations reached through the interviews would be useful. In particular when dealing with migrant organizations, given the described heterogeneity of their nature and forms, the reader should be informed on which organizations have been interviewed (key characteristics such as size, scale of operations, services offered, identity/community, etc.). It would also be important to know which criteria were adopted in making the choice of which organizations to interview.
|
Informations about the migrant organizations were added. We have described how the sample was composed and what characteristics the studied migrant organizations have. |
There are some repetitions that could be addressed to improve readability. |
We read the text again critically and found a few repetitions. We have changed a few sentences. Since most of the repetitions are rather short sentences and they each lead into a new aspect, we have not deleted all of them. |
I recommend to carefully check punctuation, brackets, capital letters, numbering of paragraphs, etc. Some editing for English language is also recommended. |
We have sent the article to a professional English-language proofreading service. |
- line 152 "Above all, the opening of Europe's internal borders, the relationship between migration and social policy have been changing fundamentally." This sentence is incomplete/unclear. - line 260 "that indicate that activities of social reproduction, which were previously performed in family contexts, have become public and thereby political affairs." It is unclear how this sentence logically follows from the one before. - line 308 "new organizations are emerge steadily" This sentence seems incorrect and therefore unclear. |
We have revised the mentioned sentences. |
Reviewer 2 Report
Contested Welfare: Migrant Organizations in search of their role in the German Welfare State
General things to consider when reviewing
• Is there a clear research issue or problem statement presented at the beginning which establishes the ‘so what’ factor?
There is a clear research question. However, the objective needs to be written in a clear and objective way; that is, it needs to be improved (“tuned” / rewritten). The objective must be clearly presented and normally appears written in the Abstract, in the Introduction and at the beginning of the conclusion. In this way, a “continuous line of thought” is built, helping the reader to understand it.
• Is the theoretical, methodological, or empirical contribution of the manuscript clearly stated? And is the significance of this contribution clearly stated?
There is “a guiding thread” throughout the paper. The text is easy to read. There is a good literature review, the methodology is clearly presented, the qualitative study is well demonstrated and well interpreted. The conclusion needs to be improved (further deepened), mentioning the main contributions of the paper to society and to future works. The authors must mention which future works would be important to do in this area of knowledge.
• Is the manuscript interesting, bold and/or innovative?
The paper is well structured. It presents a current theme, in an innovative perspective.
• Is the theoretical framework robust, providing a good conceptual grounding in relevant literature?
The bibliography is extensive, current and from reference journals.
• Is the methodology designed and executed in a reliable and valid way? Is the manuscript written in a clear and concise way (without ‘academese’) and accessible to academic and non-academic audiences?
This is a paper that uses a qualitative approach, where the technique that was used is written in a clear, concise and objective way. It is an important theme and work, both for the academic community and for people in general.
• Is the argument written in an easy to follow and logical way?
The paper is clearly written, easy to read and easy to understand and interpret. On line 51, the citation of the authors is badly done.
• Are there clear conceptual and practical conclusions drawn on in the latter parts of the manuscript?
The conclusions agree and follow on from the previous text, however the reader needs to be reminded, at the beginning of the conclusion, what was the objective of the work. It is important to deepen the conclusions further. At the end of the conclusion, authors should also mention the paper's main contributions to society and future work. The authors must mention which future works would be important to do, in this area of knowledge, to give continuity to the present work.
• Does the manuscript present an analysis of contemporary issues?
The subject of welfare state policies is current, and interesting. In the contemporary world, social support policies are a relevant idea. It is important that at European level clear methodologies are created for evaluating the return on social investment.
• Does the manuscript present a balanced perspective on the power and potential of events for good or for bad?
The paper is written in a clear, thoughtful, and balanced way. The discussion of the results is presented in a clear and objective way. The conclusions are presented in a clear and objective way but need to be improved. The conclusion lacks reminding the reader of the objective of the work and writing about possible future investigations. The bibliography is extensive and current.
• Do you think this manuscript helps advance research on the topic, how and why?
The manuscript helps advance research and knowledge on the importance of state welfare policies, on the importance of social organizations, and on the return on social investment, as well as how this theme can be increased through different types of connections between different social actors.
• Are there clear and well-justified recommendations to help advance the policy and practice of events in future?
Recommendations and guidelines on future work to be developed need to be increased.
• Does the manuscript present future academic research about the subject?
The authors hope that more research will be conducted on the theoretical advances and contributions regarding the impact of their study, however this aspect is vague. The authors must clearly and objectively identify which studies, in their opinion, should follow on from their present study.
Author Response
We would like to express our gratitude for the thoughtful and appreciative feedback. In the following, we explicate the amendments we have made with reference to the reviews. Overall, our article has gone through an English proofreading process. In addition, we address all reviewer comments and revision notes in the table below.
REVIEW 2 |
|
Revision recommendation |
Revision |
There is a clear research question. However, the objective needs to be written in a clear and objective way; that is, it needs to be improved (“tuned” / rewritten). The objective must be clearly presented and normally appears written in the Abstract, in the Introduction and at the beginning of the conclusion. In this way, a “continuous line of thought” is built, helping the reader to understand it.
|
We have made the concern clear again in the closing section and also reviewed the transitions in the article. |
The conclusion needs to be improved (further deepened), mentioning the main contributions of the paper to society and to future works. The authors must mention which future works would be important to do in this area of knowledge.
|
We have added two aspects that could/should be investigated in the future.
|
On line 51, the citation of the authors is badly done.
|
We did not find the incorrect citation in line 51. |